# SOX4 exerts contrasting regulatory effects on labor-associated gene promoters in myometrial cells

Nawrah Khader[1], Virlana M. Shchuka[1], Anna Dorogin[2,3], Oksana Shynlova[2,3,4], Jennifer A. Mitchell[1,5]*

1 Department of Cell and Systems Biology, University of Toronto, Toronto, ON, Canada, 2 Lunenfeld Tanenbaum Research Institute, Sinai Health System, Toronto, ON, Canada, 3 Department of Obstetrics and Gynaecology, University of Toronto, Toronto, ON, Canada, 4 Department of Physiology, University of Toronto, Toronto, ON, Canada, 5 Department of Laboratory Medicine and Pathobiology, University of Toronto, Toronto, ON, Canada

* ja.mitchell@utoronto.ca

**Data Availability Statement:** All relevant data are within the manuscript and its Supporting Information files.

## Abstract

The uterine muscular layer, or myometrium, undergoes profound changes in global gene expression during its progression from a quiescent state during pregnancy to a contractile state at the onset of labor. In this study, we investigate the role of SOX family transcription factors in myometrial cells and provide evidence for the role of SOX4 in regulating labor-associated genes. We show that *Sox4* has elevated expression in the murine myometrium during a term laboring process and in two mouse models of preterm labor. Additionally, SOX4 differentially affects labor-associated gene promoter activity in cooperation with activator protein 1 (AP-1) dimers. SOX4 exerted no effect on the *Gja1* promoter; a JUND-specific activation effect at the *Fos* promoter; a positive activation effect on the *Mmp11* promoter with the AP-1 dimers; and surprisingly, we noted that the reporter expression of the *Ptgs2* promoter in the presence of JUND and FOSL2 was repressed by the addition of SOX4. Our data indicate SOX4 may play a diverse role in regulating gene expression in the laboring myometrium in cooperation with AP-1 factors. This study enhances our current understanding of the regulatory network that governs the transcriptional changes associated with the onset of labor and highlights a new molecular player that may contribute to the labor transcriptional program.

## Introduction

The myometrium, or muscular tissue layer of the uterus, undergoes phenotypic changes in its progression through the process of gestation. In the pregnant, non-labor state, smooth muscle cells (SMCs) within the myometrium maintain a state of contractile quiescence. At the onset of labor, however, SMCs switch to a highly contractile phenotype and can thereby generate the required amount of force needed to deliver the fetus through synchronous contractions. Molecular changes within SMCs increase their ability to communicate with one another,

**Funding:** This work was supported by the Canadian Institutes of Health Research (FRN 173252, held by J.A.M. and O.S.; cihr-irsc.gc.ca). Studentship funding was provided by Ontario Graduate Scholarship (OGS held by N.K.). The funders had no role in study design, data collection and analysis, decision to publish, or preparation of the manuscript.

**Competing interests:** The authors have declared that no competing interests exist.

which has been shown to occur in response to increased gene and protein expression of labor-associated transcription factors. RNA sequencing (RNA-seq) data generated from myometrium tissues in human and rodent gestational models indicate that large-scale changes in mRNA levels occur alongside the myometrium transition from relative dormancy to contractility.

The phenotypic switch of the myometrium from a relatively quiescent to a highly contractile muscle stems from a coordinated interplay between hormonal, mechanical, and inflammatory signals. When dysregulated, this interplay can lead to the premature onset of labor, which in humans is associated with increased fetal mortality and morbidity [1]. Mouse models allow us to study the processes that initiate labor and how these vary between different modes of preterm labor induction. In these models, preterm labor is caused either by inhibiting progesterone receptor (PR) activity and subsequently incurring premature progesterone withdrawal using RU486 (mifepristone), or by an infection-simulation inflammation pathway via lipopolysaccharide (LPS) injection in mice. Many genes that are differentially regulated between late pregnancy and term labor are also differentially regulated at the onset of preterm labor in these mouse models. However, RNA-seq analysis has also revealed that the SMC transcriptome differs depending on the type of labor induction, with a specific set of gene expression patterns unique to each induction mode [2]. Our own work lends further evidence for transcription-based control in the onset of labor. Through total RNA-seq analyses in mouse myometrium tissues, we uncovered an increase in active transcription at labor-associated genes from late gestation to active labor [3]. We have shown that most of the genes that are upregulated at labor onset display increased primary transcript production, indicating increased transcription; however, the mechanisms that control these transcription changes remain poorly characterized.

Through multiple studies in established cell models, we know that extensive networks of transcription factors are responsible for modulating gene transcription in specific cellular contexts. These factors usually bind common regulatory elements in large protein complexes [4–6]. Since the onset of labor is controlled through tightly regulated transcription in the myometrium, a thorough understanding of the network of transcription factors that coordinate these regulatory events is warranted. Several studies have identified crucial transcription factor families as classical regulators of the labor transcriptional program—such as activator protein 1 (AP-1), nuclear factor kappa beta (NF-κβ), estrogen receptor (ER), and PRs; recent studies have also highlighted transcription factors belonging to the KLF, SMAD, FOXO, IRF, and HOXA families as potential cooperative candidates driving labor-associated gene expression as part of a labor-specific transcriptional regulatory network (reviewed in [7]). Recently, we found that two additional factors, MYB and ELF3, exhibit differential expression in pre-laboring and laboring human and murine myometrium as SMCs undergo the quiescent-to-contractile transition. We found that despite being upregulated at labor onset, these factors play opposing roles in stimulating expression from labor-associated gene promoters, with MYB having an activating and ELF3 a repressive role [8].

To further expand the repertoire of proteins that belong to a labor-associated regulatory complex, we present evidence here for additional factors that contribute to gene regulation in uterine SMCs: members of the SRY-Box (SOX) transcription factor family. Microarray, RNA-seq data, and peptidomics analysis reveals SOX4 is upregulated in the human myometrium at term labor [9–11] compared to non-laboring samples. Several SOX family factor motifs which include, but are not limited to, SOX4, SOX9, SOX17, were also found to be uniquely enriched in promoter regions marked by an active histone modification (H3K4me3) in human laboring tissues relative to nonlaboring tissues [12]. RNA-seq data from the murine myometrium demonstrates that *Sox4*, *Sox7*, and *Sox9* are upregulated whereas *Sox8* is downregulated at labor

onset compared to late pregnancy [3]. Furthermore, the promoter of *Gja1*, a gene used as a proxy for transcription at labor, owing to its elevated expression in the laboring myometrium, [3,13–15] contains several SOX motifs. Interestingly, SOX4, SOX8, and SOX9 were also found to cooperate with cJUN, a member of the AP-1 family, in activating the expression of *Gja1* in Sertoli cells [16]. Considering these findings, we hypothesized that SOX factors may be crucial regulators of the transcriptional program in laboring SMCs.

Here, we assess the gene expression levels of *Sox4*, *Sox7*, *Sox8*, and *Sox9* throughout gestation and the onset of term labor and in preterm labor models. Based on our findings, we focused on *Sox4* which displayed increased expression as SMCs in the mouse myometrium transition from quiescence during gestation to contractility at term and preterm labor onset. In our further assessment of the role that SOX4 may play in driving the expression of key labor-promoting genes, we demonstrate that SOX4 makes context-specific contributions in regulating labor-associated gene promoter activity. We find that SOX4, either on its own or with AP-1 factors, does not regulate *Gja1* expression. However, SOX4 contributes to the activation of both the *Fos* and *Mmp11* promoters, in cooperation with AP-1 factors. Interestingly, our results show that SOX4 has a repressive effect on the *Ptgs2* promoter, both on its own and in combination with AP-1 dimers. Through our findings, we demonstrate the diverse regulatory role that SOX4 plays in modulating gene expression while expanding the list of candidate transcription factors that can cooperatively control the timing of labor.

## Results

### Specific members of the SOX transcription factor family are upregulated at labor onset in the murine myometrium

To examine the expression levels of *Sox4*, *Sox7*, *Sox8*, and *Sox9* across a larger set of gestational timepoints, we performed RT-qPCR on RNA samples extracted from murine myometrium tissues at various points across the gestational profile, including gestational day 15 (D15), day 17 (D17), term-not-in-labor (D19; TNIL), term labor (D19.5; LAB), and 1 day postpartum (Day 20; 1PP). Across the gestational timeline, we observed significantly increased expression at term (TNIL) relative to late gestation for *Sox4* [12-fold increase], and *Sox7* [14-fold increase] (Fig 1). *Sox9* gene expression was significantly elevated at TNIL compared to late gestation, but not at LAB (S1 Fig in S1 File). Interestingly, the expression of *Sox8* was not significantly different between any of the tested gestational timepoints compared to labor despite being significantly downregulated towards labor onset in the RNA-seq data (S2 Fig in S1 File). The increase in transcript expression across most of the tested SOX transcription factor-encoding genes occurs immediately prior to the onset of labor, implicating these factors as modulators of labor in the mouse myometrium.

### *Sox4* is associated with premature labor onset in both progesterone withdrawal- and infection-driven inflammation-based mouse models of preterm labor

Given that *Sox4*, *Sox7*, and *Sox9* may be key regulators of the laboring SMC phenotype, we next sought to determine whether these factors exhibit similar elevated expression trends in mouse models of preterm labor. We used two well-established models of mouse preterm labor: (1) a model using RU486, or mifepristone, which causes the premature withdrawal of progesterone and induces preterm labor; and (2) an inflammation-based model via injection of *E. coli*-derived lipopolysaccharide (LPS), which prompts uterine inflammation and consequent preterm labor. In the progesterone withdrawal preterm labor model, we observed that the

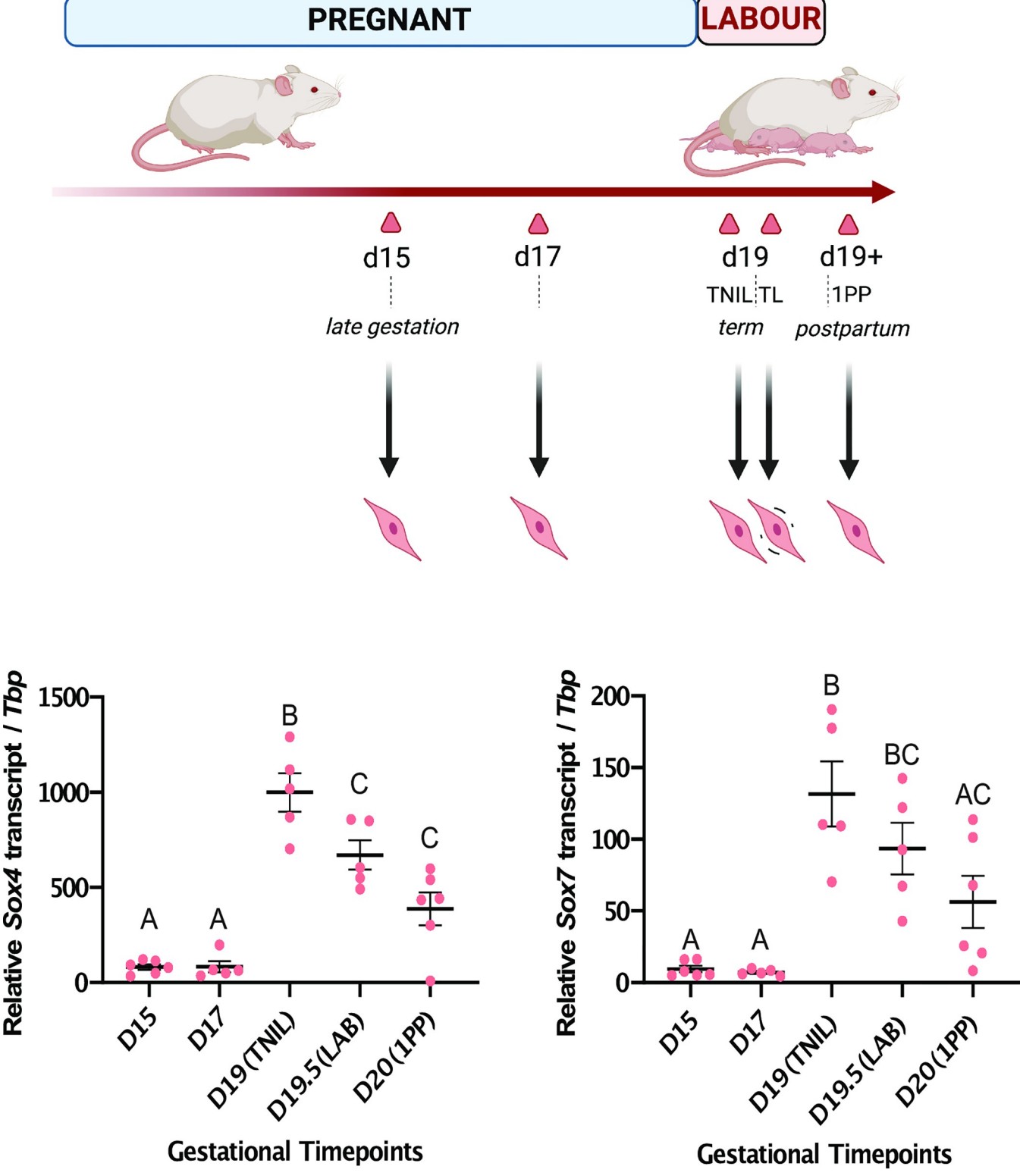

**Fig 1.** *Sox4* **and** *Sox7* **genes exhibit elevated expression levels in term labouring murine myometrium tissues relative to gestation.** (A) Schematic of mouse gestational timepoints when myometrium tissues were collected. Gestational timepoints indicated either by day (D) or status [term-not-in-labor (TNIL), labor (LAB), and postpartum (PP)] are marked by triangles, alongside respective associated gestational stages. (B) Transcript expression levels of *Sox4* and *Sox7* genes were measured by RT-qPCR (±SEM) at indicated timepoints. Groups that exhibit significant differences (p < 0.05) as determined by one-way ANOVA are distinguished by different letters, while groups that do not show significant differences (p > 0.05) are labeled with the same letter.

transcript levels of *Sox4* and *Sox7*, were significantly elevated (downregulated for *Sox8*) at labor relative to vehicle-treated mice on the same day (Fig 2A and S3B Fig in S1 File). Contrarily, upon assessing expression profiles of the transcription factor-encoding genes of interest in the inflammation-based model, we observed that only *Sox4* is significantly upregulated at labor compared to sham-treated mice on the same day (Fig 2B). *Sox9* levels displayed no change in either preterm labor model (S3B Fig in S1 File). We were surprised to observe that, of these factors, *Sox7* (upregulated) and *Sox8* (downregulated) show differential expression only in RU486-induced preterm labor. These data suggest that SOX7 and SOX8 may be regulated by progesterone signaling pathways.

## SOX4 protein increases in abundance at labor onset in a gestational context

Since we observed that only *Sox4* displayed significantly elevated transcript expression in term labor and both preterm labor mouse models, we next asked whether this increase in expression at labor onset is also reflected at the protein level. Using immunoblot analysis, we investigated levels of SOX4 in mouse myometrium tissue samples obtained from nonpregnant mice (NP) and pregnant mice at various timepoints throughout the course of gestation (Days 8, 12, 15, 17, 19), as well as at labor (Day 19.5; LAB), and postpartum (Day 20; 1PP) [Fig 3A]. Densitometric analysis revealed that SOX4 protein abundance is significantly higher by a factor of 4-fold at labor compared to nonpregnant myometrium, as well as at gestational days 8 and 12 timepoints (Fig 3C; representative immunoblot shown in Fig 3B and S4 Fig in S1 File). These data lend further support for the role that SOX4 may be playing in the laboring myometrium.

## SOX4 drives *Fos* but not *Gja1* promoter activation in cooperation with AP-1 homodimers

Having established that murine myometrium display increased levels of SOX4 protein at the onset of both term and preterm labor, we next sought to ascertain whether SOX4 exerts a functional role in the transcriptional control of labor-associated genes which contain strong binding motifs for both AP-1 and SOX4. We used a luciferase reporter assay as a means of measuring promoter regulation in transfected Syrian hamster myometrium (SHM) cells [8,13,17]. We used previously generated luciferase constructs with the reporter gene under the control of the well-studied labor-associated promoters from the *Gja1* and *Fos* genes [8] to assess the regulatory capacity of SOX4, either on its own or alongside AP-1 factors. We chose to focus on FOSL2 and JUND within the AP-1 FOS and JUN subfamilies due to their significant protein expression during labor onset in mice [18]. These factors were thus selected for our reporter assay investigations using AP-1 homodimer and heterodimer combinations. We assessed whether the transactivation potential of the AP-1 dimer combinations on the *Gja1* and *Fos* gene promoters is altered in the presence of SOX4 (Fig 4A and 4B). Note that the two controls (*Gja1* and *Fos* promoter activities indicated by asterisks in Fig 4A and 4B) increased with the addition of JUND, and were significantly elevated in the presence of JUND and FOSL2 proteins, as previously reported [8]. Here we show that the addition of SOX4 does not significantly activate the *Gja1* promoter, either on its own or in association with the AP-1 proteins, despite the presence of a strong SOX4 motif in the promoter sequence (Fig 4A). In the case of the *Fos* promoter, while we observed no significantly increased activity of the *Fos* promoter in the presence of SOX4 on its own or alongside JUND/FOSL2 AP-1 heterodimers, SOX4 presence incurred a significant activation effect in the presence of JUND homodimers (Fig 4B). In light of these results, SOX4 appears have promoter context-specific regulatory effects and, in the case of the *Fos* gene promoter, works in cooperation with JUND.

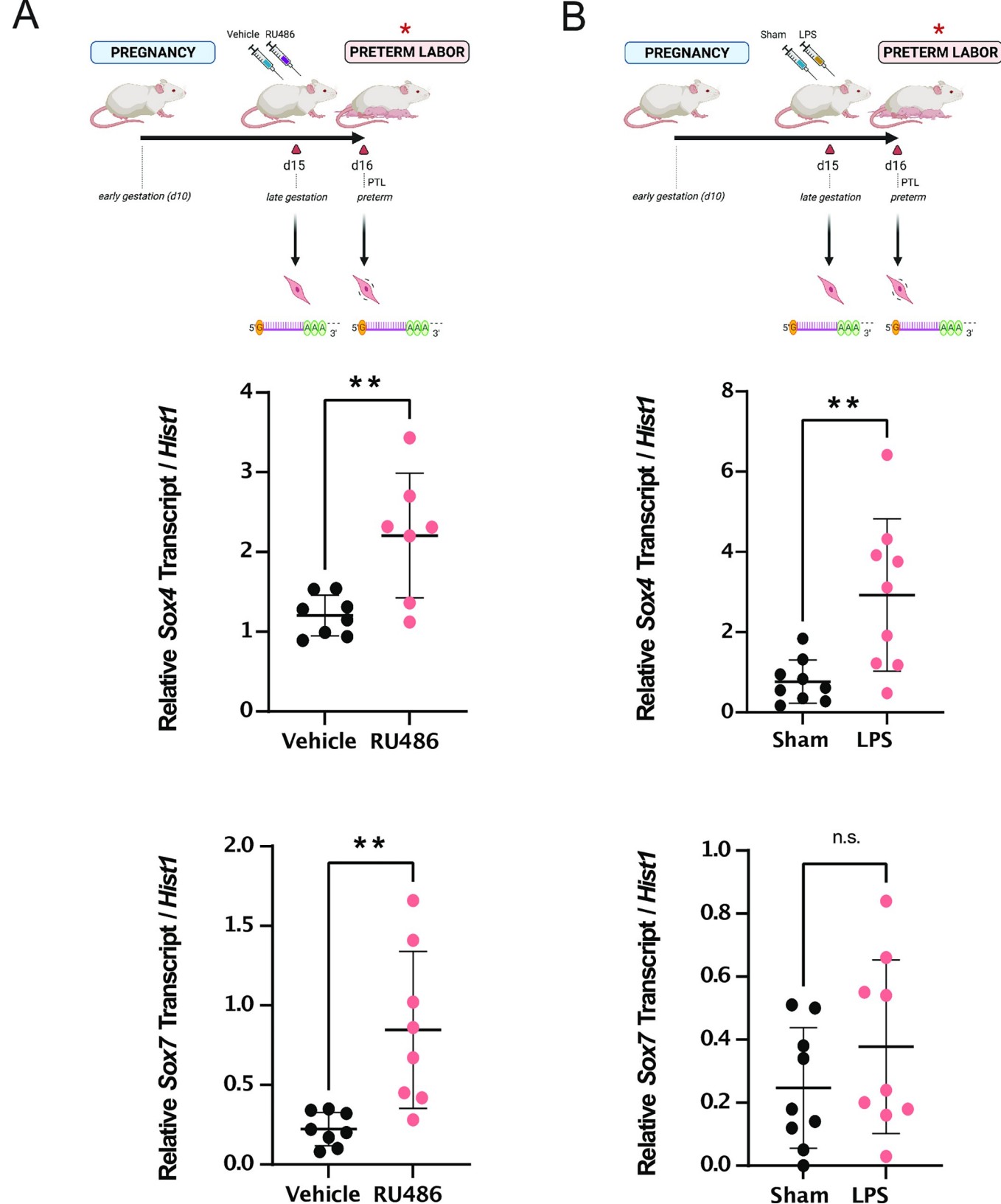

**Fig 2. *Sox4* is upregulated during the onset of RU486- and LPS-induced preterm labor and *Sox7* is upregulated specifically in the RU486-induced preterm labor model.** (A) Schematic of mouse gestational timepoints when myometrium tissues were collected in two models of preterm labor induction: loss of progesterone (RU486) and local infection simulation (LPS). Overview of timing of RU486 (*left*) or LPS *(right)* treatments and subsequent labor events relative to respective sham and vehicle controls, as indicated by triangles. (B) Transcript expression levels of *Sox4* and *Sox7* genes were measured by RT-qPCR (±SEM) at the indicated timepoints. Differences in expression levels between control and preterm labor that are statistically significant are marked by ** ($p < 0.01$).

## The *Mmp11* promoter is activated in the presence of SOX4 and AP-1 factors

After assessing the regulatory capability of SOX4 on the *Gja1* and *Fos* promoters, we further investigated its potential effects on other labor-associated gene promoters. We generated another reporter construct under the control of the *Mmp11* promoter sequence as it was found to contain strong binding motifs for SOX4 and AP-1. Encoding a matrix metalloproteinase 11 (also known as stromelysin-3), *Mmp11* has been shown to be upregulated in the rodent myometrium at the onset of labor and sustained throughout the postpartum period relative to gestation [8,19]. MMP11 is thought to contribute to changes in extracellular matrix composition and the basement membrane surrounding SMCS, in preparation for coordinated contractions at the onset of labor. We first tested the effects of either JUND or JUND and FOSL2 on *Mmp11* promoter-mediated reporter gene expression. Surprisingly, despite having two AP-1 binding sites [20], our results show that neither JUND alone nor JUND and FOSL2 together had any significant effect on the transcriptional output of the *Mmp11* promoter (Fig 4C). We next turned to test how the presence of SOX4 might affect *Mmp11* promoter output. On its own, SOX4 did not significantly modulate the *Mmp11* reporter expression, but we found that, in combination with either JUND: JUND homodimers or JUND: FOSL2 heterodimers, SOX4 presence increased the *Mmp11* promoter activity by a factor of 4-fold (Fig 4C).

## The *Ptgs2* promoter is positively regulated by AP-1 factors, but negatively regulated upon addition of SOX4

Upon establishing that SOX4 imposes different effects on the *Gja1*, *Fos*, and *Mmp11* promoters, we applied this assay to another well-established labor-associated gene, *Ptgs2* [encoding Prostaglandin synthase 2, also known as COX2] which also contains both SOX4 and AP-1 binding motifs. *Ptgs2* is a well-studied factor in the context of labor, one involved in synthesizing the uterotonic agents, prostaglandins, which in turn stimulate contractions within SMCs in the laboring myometrium [21–23]. Additionally, the expression of *Ptgs2* is elevated at the onset of labor in both human and murine contexts [3,10,24–26]. Prior to our testing the effects of SOX4 on the *Ptgs2* gene promoter, we were interested in assessing how either JUND or JUND and FOSL2 together might regulate this promoter, as this had not been previously determined. Therefore, we tested the effects of overexpression of JUND or JUND and FOSL2 together on expression levels of the reporter gene under the control of the *Ptgs2* promoter. We found that, similar to the case of the *Fos* and *Gja1* promoters, the addition of JUND demonstrated a minimal elevation of reporter expression levels compared to reporter activity in the absence of any factor. Furthermore, we observed a significant 5-fold upregulation by JUND and FOSL2 together on *Ptgs2* promoter-mediated reporter expression levels (Fig 4D). However, the addition of SOX4 alone did not significantly activate the expression of the *Ptgs2* promoter, and surprisingly downregulated *Ptgs2* promoter-mediated reporter expression in combination with JUND and FOSL2, compared to the heterodimer on its own (Fig 4D, *right*). These findings suggest that while AP-1 factors positively regulate the expression of *Ptgs2*, the presence of SOX4 overrides the activation effect of JUND and FOSL2 and represses *Ptgs2* transcription in SHM cells.

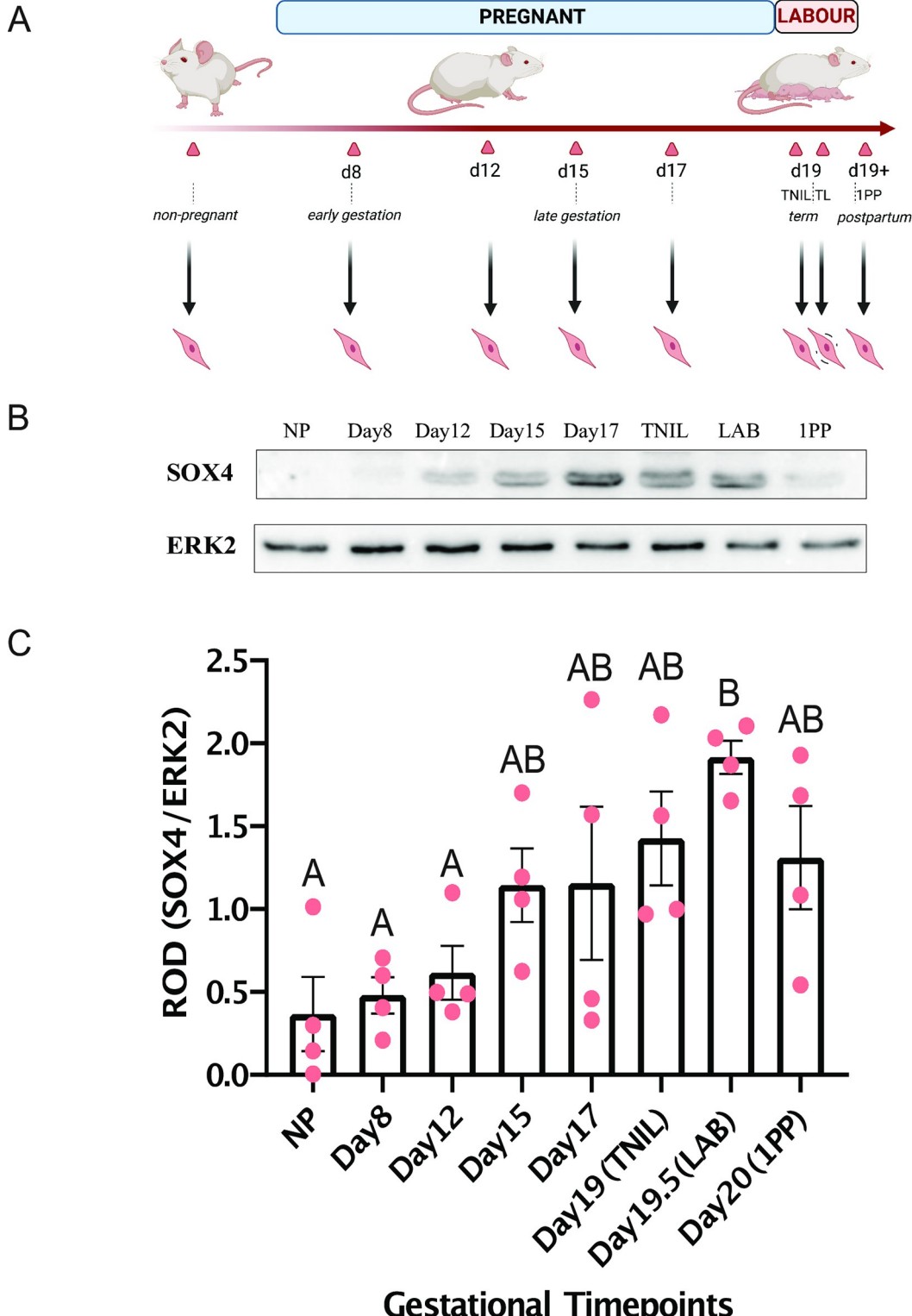

**Fig 3. SOX4 protein expression is significantly elevated in laboring murine myometrium tissues relative to gestation.**
(A) Schematic of mouse gestational timepoints when myometrium tissues were collected. Gestational timepoints indicated either by day (D) or status [term-not-in-labor (TNIL), labor (LAB), and postpartum (PP)] are marked by triangles, alongside respective associated gestational stages. (B) Representative immunoblot image of expression of SOX4 and housekeeping protein, ERK2, is shown at the indicated timepoints. (C) Densitometric analysis (±SEM) of SOX4 protein

lysates extracted from mouse myometria at various gestational stages. Groups that exhibit significant differences (p < 0.05) as determined by one-way ANOVA are distinguished by different letters, while groups that do not show significant differences (p > 0.05) are labeled with the same letter.

## Discussion

Our work reveals a regulatory role for SOX factors in the laboring transcriptional program and identified *Sox4* as the SOX factor most consistently upregulated at labor onset, both in term and preterm labor contexts. Through our gene expression analysis, we also observed that despite being elevated in term labor mouse models, *Sox7* was only significantly upregulated in a preterm mouse model which involves the loss of progesterone signalling. Similarly, *Sox8* was also only significantly downregulated when preterm labor was induced via the PR antagonist, RU486. These data suggest that SOX7 and SOX8 may be involved in the onset of labor only when progesterone signalling is interrupted either at term or preterm. This is consistent with observations from previous studies where these factors were shown to be affected by progesterone signalling [27]. Notably, Yin et al. found that when human uterine leiomyoma SMCs were exposed to RU486, *SOX8* mRNA levels were significantly downregulated [27]. RNA-seq analyses using the RU486 preterm labor mouse model also showed downregulation of *Sox8* toward preterm labor [2]. Furthermore, an enrichment of SOX motifs in PR ChIP-seq datasets conducted in the uterus supports the hypothesis that SOX factors and PRs may cooperatively regulate uterine function [28]. Our gene expression findings suggest that *Sox7* and *Sox8* may differentially contribute specifically to a progesterone withdrawal-driven laboring phenotype in the case where the recruitment of SOX7 and the loss of SOX8 causes the onset of labour.

SOX4, our data finds, could play a crucial role in modulating the phenotypic switch of SMCs from relative quiescence to contractility as it was the only factor that was significantly upregulated at term and in both preterm labor models. Indeed, SOX4 has been shown to be elevated in the laboring mouse myometrium through microarray and RNAseq analysis [3,9] and in the human myometrium as demonstrated via peptidomics analysis [11] relative to the respective nonlaboring tissues. As demonstrated by our results, the addition of SOX4 either on its own or in cooperation with AP-1 factors had regulatory activity at the promoters of known labour associated genes; however, SOX4 did not regulate the *Gja1* promoter. This result was surprising at first because Ghouili et al. showed SOX and JUN factors can transactivate their targets together; however, our result is consistent with Najih et al. who demonstrated the dispensable role of SOX4 on *Gja1* promoter activity in spermatogonia GC-1 cells [16,29]. Contrarily, SOX4 demonstrated a notable activating effect on the *Fos* promoter, but only in a cellular context that lacks FOSL2 protein and overexpresses JUND. One possible explanation for this result could involve binding of SOX4 with JUND homodimers to regulate the expression of *Fos* in the absence of FOS proteins; however, in a cellular context that includes FOS protein members like FOSL2, JUND preferentially binds with FOSL2 and exerts a more substantial transcriptional activation effect on its target genes. Although, SOX4 had no activating capacity at the *Gja1* promoter, the ability of SOX4 in conjunction with JUND, to increase transcription driven by the *Fos* promoter can indirectly increase *Gja1* expression as FOS is a more potent activator of *Gja1* promoter activity than JUN proteins alone [13,30]. The nature of the reporter assay, however, is such that it does not perfectly recapitulate the cellular environment of SMCs undergoing labor, and therefore does not account for additional components like hormone and mechanical signals that may modify how proteins behave in a labor context. Despite these results, our findings are important in highlighting the role of SOX4 in regulating transcription in myometrial cells.

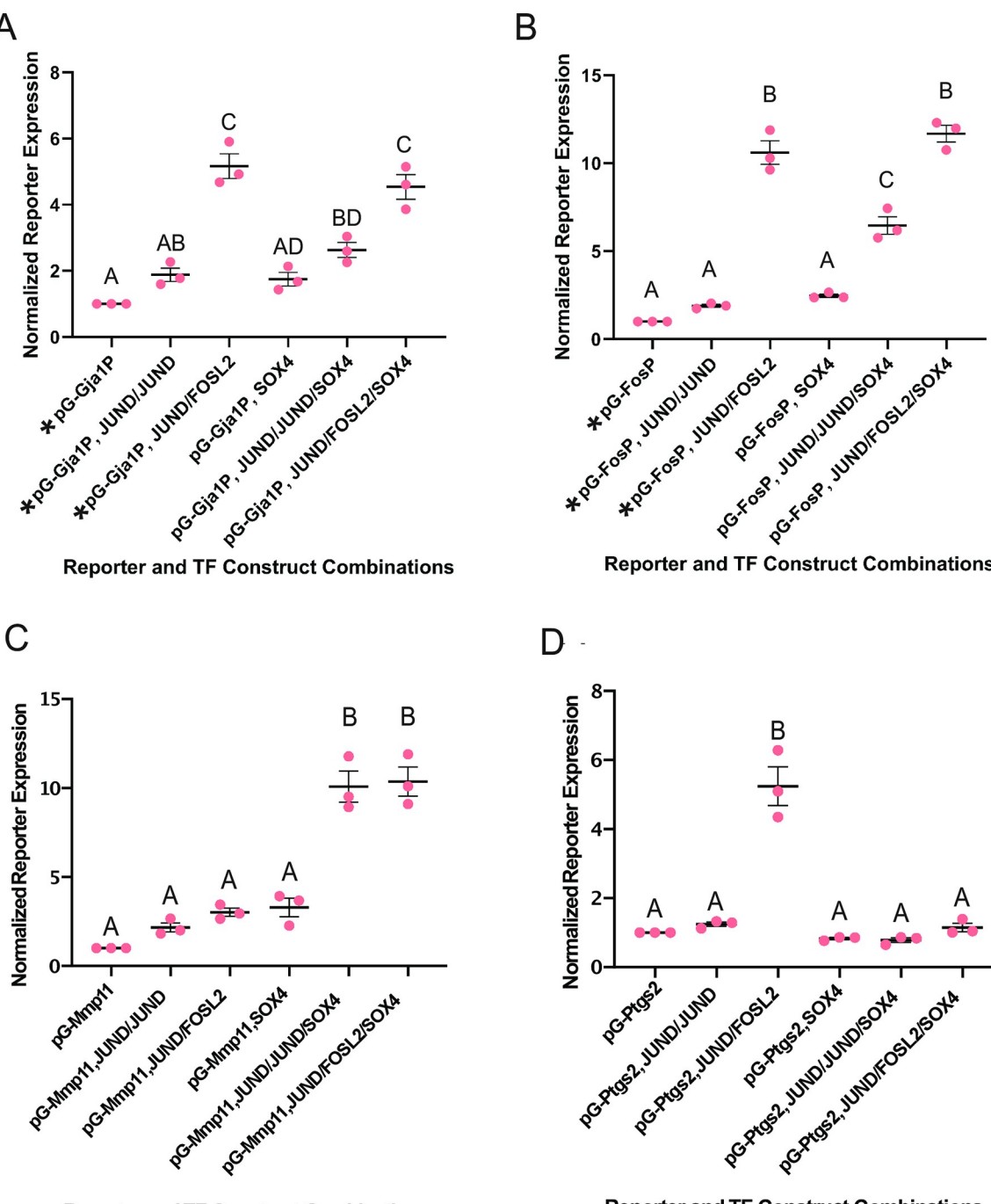

**Fig 4. SOX4 differentially alters the promoter activating effects of AP-1 factors on the promoters of *Fos*, *Mmp11*, and *Ptgs2*, but not *Gja1*.** The relative expression levels of reporter genes driven by the mouse *Gja1* (A), *Fos* (B), *Mmp11* (C) or *Ptgs2* (D) promoters from luciferase constructs compared to expression output of Renilla constructs. Output comparison among addition of JUND: JUND, JUND: FOSL2, SOX4 alone, SOX4 and JUND: JUND homodimers, and SOX4 and JUND: FOSL2 heterodimers relative to no transcription factor addition. Average expression level values (±S.D.) for each construct are normalised to the expression output of the promoter construct without TF treatment. Significantly different results (p<0.05) are indicated by groups labeled with different letters. Asterisks (*) in (A) and (B) indicate data from control groups reprinted from [8]. The data for additional groups in (A) and (B) that contain SOX4 were generated alongside control group data.

Compared to our findings from the *Fos* and *Gja1* promoters, we noticed contrasting trends in the AP-1 mediated *trans*-activation of the *Mmp11* and *Ptgs2* promoters with the addition of SOX4. Unlike the *Fos* promoter, where SOX4 seemed to have a context-dependent effect, our data suggests that SOX4 may act synergistically with either AP-1 homo- or heterodimers (fold change > 1.5 over the additive fold change of AP-1 dimers and SOX4 alone) in activating reporter-driven gene expression at the *Mmp11* promoter. To further assess whether SOX4 acts in synergy with AP-1 factors, additional protein-protein interaction tests, such as proximity ligation assays or co-immunoprecipitation experiments are warranted. Surprisingly, the addition of SOX4 to AP-1 factors in the *Ptgs2* promoter exhibited a repressive effect. Previously, we found that, like SOX4, ELF3 exhibited a spike in expression at labor, but a repressive regulatory effect on the *Gja1* and *Fos* promoters [8]. These opposing activation effects seen through our data suggests that, like ELF3, SOX4 imparts both an activating and repressing effect within SMCs in regulating labor-associated gene expression. Taken together these findings reveal the nuances through which transcription factor complexes can coordinately control both the levels and timing of gene expression changes in a complex process such as the onset of labor.

Ascertaining the exact regulatory role that SOX4 exerts on specific labor-associated genes would require further investigation as SOX4 appears to perform a dynamic role within the SMC context. Similar to myometrial cells, SOX4 has shown to exhibit both transcriptional activator and repressor capacities in various cancerous cell contexts [31,32] depending on its posttranslational modifications or interaction with additional factors. Because of the ability of SOX4 to activate or repress its target genes, SOX4 may contribute to differential regulation of labor-associated genes, depending on how the SOX4 protein is modified at the posttranslational level, other factors SOX4 interacts with, and/or the predominance of labor-triggering signals. Indeed, SOX4 was shown to be recruited at the promoter of a labor-driving gene, *Cald1*, during differentiation of muscle cells upon being acetylated [33]. SOX4 has also been shown to interact with other transcription factors to regulate gene expression. For example, in fibroblast-like synoviocyte cells, SOX4 interacts with RELA to coregulate genes involved in pathways similar to those enriched at labor, such as inflammation and AP-1 signaling pathways [34]. RELA is a transcription factor from the NF-kB family that has been shown to be expressed in the laboring myometrium [34–36]. Finally, hormonal signaling was also shown to influence the regulation and expression of SOX4. Chromatin immunoprecipitation assays have confirmed the binding of estrogen receptor B to the *SOX4* gene promoter in prostate adenocarcinoma cells (LNCaP), suggesting a regulation of SOX4 through estrogen signaling, a prominent process in controlling the timing of labor onset in the myometrium (reviewed in [7,37]). This is in agreement with Hunt and Clarke, (1999), who have shown *Sox4* mRNA expression levels differ in mouse uterine tissues in response to changes in estrogen levels [38]. To further delineate the exact regulatory role that SOX4 plays in the myometrium, chromatin immunoprecipitation experiments using non-laboring and laboring SMCs would allow for an assessment of whether SOX4 binds and could regulate labor-promoting genes.

Taken together, this work expands the current repertoire of factors that contribute to a laboring transcriptional phenotype and provides support for a previously understudied factor in the context of labor. We have demonstrated a role for SOX4 in modulating labor-associated gene expression in SMCs. Our work sets a foundation for further research regarding the cooperation of SOX4 within the broader transcriptional machinery that regulates the timing of labor. Developing this deeper understanding of the molecular foundation of gestation and labor can uncover an extensive array of potential treatment targets and facilitate an informed development of measures for the prevention of premature labor in women.

## Methods

### Animal model

All animal experiments were approved by the Animal Care Committee of The Centre for Phenogenomics (TCP) (Animal Use Protocol #0164H). Guidelines set by the Canadian Council for Animal Care were strictly followed for handling of mice. Virgin outbred CD-1 in these experiments were purchased from Harlan Laboratories (http://www.harlan.com/). All animals were housed in a pathogen-free, humidity-controlled 12h light, 12h dark cycle TCP facility with free access to food and water. Female mice were naturally bred; the morning of vaginal plug detection was designated as gestational day (D) 1. For CD-1 mice gestation length is 19.5 days, spontaneous delivery occurred during the evening of D19 or morning of D20.

### Mouse term labor model

In the spontaneous term labor gestation model, samples were collected on gestational days 15, 17, 19 term-not-in-labor (TNIL), 19–20 during active labor (LAB), and within 24 hours postpartum (pp) for CD-1 mice. Myometrium tissues were collected at 10 AM on all gestational days with the exception of labor samples (LAB), which were collected upon the delivery of at least one pup.

### RU486-induced mouse preterm labor model

On day 15 of gestation, mice were treated with either RU486 (150 μg in 100 μl corn oil containing 10% EtOH, 17ß-hydroxy-11b-[4-dimethylaminophenyl]-17-[1-propynyl]-estra-4,10-dien-3-ne; mifepristone) or with vehicle (oil/ethanol solvent). Preterm labor occurred 24 ± 2 hour after RU486 injection. Myometrium tissue samples were collected during active labor after delivery of at least one pup. In vehicle-treated mice, samples were collected 24 hours after injection of the vehicle solution.

### Lipopolysaccharide (LPS)-induced mouse preterm labor model

On day 15 of gestation, mice underwent a mini-laparotomy and given an intrauterine infusion of lipopolysaccharide (LPS; 125 μg in 100 μL of sterile saline) between two lower amniotic sacs or administered 100 μL of sterile saline (sham). The LPS used for this study was isolated from *E. coli* serotype 055: B5. Mice were sacrificed during LPS-induced preterm labor, occurring 12–24 hours post LPS infusion. In mice undergoing sham surgery, samples were collected 24 hours after surgery.

### Myometrium tissue collection

Prior to tissue collection at the appropriate timepoint, mice were euthanized by carbon dioxide inhalation. The part of the uterine horn close to the cervix, from which the fetus was already expelled during term or preterm labor, was removed; the remainder was collected for analysis. Both uterine horns were bisected longitudinally and dissected away from both pups and placentas and placed in ice-cold PBS. Through mechanical scraping on ice, the decidua, as well as the uterine stromal tissues, were carefully removed from the myometrium tissue. Myometrium tissues were washed in ice-cold PBS prior to flash-freezing in liquid nitrogen and subsequently stored at −80˚C until needed.

## Gene expression quantification by RNA extraction and RT-qPCR

Total RNA was extracted from frozen, crushed myometrium tissue samples using a Trizol/Chloroform extraction protocol. RNA samples were treated with DNase I (Qiagen) and were column-purified using RNeasy Mini Kit (Qiagen) to remove genomic DNA. Using a high-capacity cDNA synthesis kit (Thermo Fisher Scientific), RNA samples were reverse transcribed to cDNA. Genomic DNA was extracted using the Qiagen DNA Blood and Tissue Extraction kit and used to construct a standard curve to quantify gene expression levels. SYBR® Select solution was used to perform quantitative PCR on cDNA samples to determine gene expression levels. All samples were confirmed not to have DNA contamination via reverse transcriptase negative samples. Target gene expression levels were normalised to a reference gene (*Tbp* or *Hist1*) using gene-specific primers that amplify sequences within the exon regions (S1 Table in S1 File).

## Syrian hamster myometrium (SHM) cell culture

Syrian hamster myometrium (SHM) cells, generously supplied by Professor Oksana Shynlova, were cultured on plates in phenol red-free DMEM media supplemented with 10% FBS and 1% penicillin-streptomycin (Pen-Strep) and kept in a 37°C/5% $CO_2$ environment. Cells were passaged and/or had their media replaced, as appropriate, every 2–4 days. Cell media was tested and confirmed to be negative for presence of mycoplasma.

## Reporter plasmid acquisition and cloning

Luciferase constructs were generated as previously described in [8]. Briefly, promoter regions were amplified from Bl6 murine genomic DNA. The original minimal promoter from the pGL4.23 construct (Promega) was first removed via digestion with BglII and NcoI and the plasmid backbone was purified. Labor-associated endogenous gene promoters were cloned into the pGL4.23 backbone either directly from gDNA (using overhang-containing primers) in the case of the *Fos* promoter (Addgene catalog #188113), or from transitional pJET-1.2 vector backbones containing the cloned promoter, in the case of the *Gja1* (Addgene catalog #188114), *Ptgs2* (Addgene catalog #209519), and *Mmp11* (Addgene catalog #209520) promoters (**S2** Table in S1 File). Generation of FOSL2- and JUND-encoding plasmids is described in [13] and can be obtained from Addgene (catalog #187907 and #187904, respectively). The pcDNA3-FLAG-SOX4 was a gift from Carlos Moreno (Addgene plasmid # 110360; http://n2t.net/addgene:110360; RRID:Addgene_110360) [39]. Expression of these constructs was confirmed by western blot (see S2 Fig in [8]).

## Western blot

Western blot analysis was conducted as previously described [17]. Briefly, nuclear proteins were extracted from myometrium tissue samples using a Nuclear and Cytoplasmic Extraction kit (Pierce, USA) and lysis buffer with freshly added protease and phosphatase inhibitor cocktail was added to protein lysates. A BCA protein assay kit (Pierce) was used to determine protein concentration and equal lysate amounts were run through an SDS-PAGE gel. Gel contents were transferred to a polyvinylidene fluoride membrane (PVDF). Blocking was performed for an hour with 5% milk in TBS-T and samples were incubated with the primary antibody (anti-SOX4: LSBio-C49999849) overnight at 4°C. Membranes were washed and probed with horseradish peroxidase (HRP)-conjugated secondary antibody at room temperature for one hour. Signals were detected using HRP substrates before imaging with BioRad Image Lab software. Membranes were then stripped and re-probed using an antibody that detects total

ERK2 (Santa Cruz, sc-154), as this protein was shown to have the most stable expression in the myometrium [18,40]. ERK2 was used to evaluate loading consistency and used for normalization across samples. All quantifications were done using ImageJ software.

## Luciferase assays

Activity of predicted labor-associated TF candidates was assayed using a dual luciferase reporter assay (Promega) as done previously [8]. On Day 0, SHM cells were seeded in antibiotic- and serum-free media at a density of $5\times10^4$ cells/well within a 24-well plate format. On Day 1, media was replaced with Opti-Mem 1 hour prior to transfection. Using Lipofectamine 3000, molar equivalents of the appropriate TF constructs were transfected into cells alongside a 1:1 molar ratio of Luciferase (pGL4.23) to Renilla (pGL4.75, Promega) vectors. This was our optimised ratio for most uniform Renilla levels across vector treatments. Total transfected DNA for any one vector combination set did not exceed a maximum amount of 500 ng/well. On Day 2, Opti-Mem media was replaced with fresh antibiotic- and serum-free SHM media. On Day 3, media was aspirated, and cells were lysed in 1X passive lysis buffer/PBS, incubated at RT on a shaker for 15 min, and subsequently frozen at −80°C for at least one hour and up to one week. Reporter activity was measured on the Fluroskan Ascent FL plate reader 48 hpt and calculated by normalising the Luciferase to *Renilla* expression.

## Statistical analyses

Significant changes in myometrium gene expression across mouse term gestational profiles were determined by one-way ANOVA with Tukey correction. Changes in gene expression in mouse preterm labor models were determined by unpaired t-tests using GraphPad Prism 9. Luciferase assay data were analysed by two-way ANOVA using GraphPad Prism 9 and significant differences were confirmed by the Holm-Sidak method.

## Supporting information

**S1 File. Supporting information for: SOX4 exerts contrasting regulatory effects on labor-associated gene promoters in myometrial cells.**
(PDF)

## Acknowledgments

We would like to thank Dr. Lubna Nadeem for her extensive guidance and instruction in the SHM cell culture and luciferase assay experiment design. Select figures in this study were created using BioRender (Biorender.com).

## Author Contributions

**Conceptualization:** Nawrah Khader, Jennifer A. Mitchell.

**Data curation:** Nawrah Khader.

**Formal analysis:** Nawrah Khader, Virlana M. Shchuka.

**Funding acquisition:** Nawrah Khader, Oksana Shynlova, Jennifer A. Mitchell.

**Investigation:** Nawrah Khader, Virlana M. Shchuka.

**Methodology:** Nawrah Khader, Virlana M. Shchuka, Anna Dorogin.

**Project administration:** Nawrah Khader, Oksana Shynlova, Jennifer A. Mitchell.

**Resources:** Oksana Shynlova, Jennifer A. Mitchell.

**Supervision:** Jennifer A. Mitchell.

**Validation:** Nawrah Khader, Virlana M. Shchuka.

**Visualization:** Nawrah Khader, Virlana M. Shchuka.

**Writing – original draft:** Nawrah Khader.

**Writing – review & editing:** Nawrah Khader, Virlana M. Shchuka, Anna Dorogin, Oksana Shynlova, Jennifer A. Mitchell.

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
