## [Decision Letter · Decision Letter 0]

22 Nov 2023

PONE-D-23-29498SOX4 exerts contrasting regulatory effects on labor-associated gene promoters in myometrial cells.PLOS ONE

Dear Dr. Mitchell,

Thank you for submitting your manuscript to PLOS ONE. After careful consideration, we feel that it has merit but does not fully meet PLOS ONE’s publication criteria as it currently stands. Therefore, we invite you to submit a revised version of the manuscript that addresses the points raised during the review process.

We look forward to receiving your revised manuscript.

Kind regards,

Atsushi Asakura, Ph.D

Academic Editor

PLOS ONE

Journal Requirements:

This work was supported by the Canadian Institutes of Health Research (FRN 173252, held by J.A.M. and O.S.; cihr-irsc.gc.ca). Studentship funding was provided by Ontario Graduate Scholarship (OGS held by N.K.). The funders had no role in study design, data collection and analysis, decision to publish, or preparation of the manuscript.

5. We note that Figures 4A and 4B in your submission contain copyrighted images. All PLOS content is published under the Creative Commons Attribution License (CC BY 4.0), which means that the manuscript, images, and Supporting Information files will be freely available online, and any third party is permitted to access, download, copy, distribute, and use these materials in any way, even commercially, with proper attribution. For more information, see our copyright guidelines: http://journals.plos.org/plosone/s/licenses-and-copyright.

We require you to either present written permission from the copyright holder to publish these figures specifically under the CC BY 4.0 license, or (2) remove the figures from your submission:

a. You may seek permission from the original copyright holder of Figures 4A and 4B to publish the content specifically under the CC BY 4.0 license. 

Reviewers' comments:

Reviewer's Responses to Questions

**Comments to the Author**

1. Is the manuscript technically sound, and do the data support the conclusions?

Reviewer #1: Partly

Reviewer #2: Partly

2. Has the statistical analysis been performed appropriately and rigorously? 

Reviewer #1: I Don't Know

Reviewer #2: Yes

3. Have the authors made all data underlying the findings in their manuscript fully available?

Reviewer #1: Yes

Reviewer #2: Yes

4. Is the manuscript presented in an intelligible fashion and written in standard English?

Reviewer #1: Yes

Reviewer #2: Yes

5. Review Comments to the Author

Reviewer #1: SOX4 exerts contrasting regulatory effects on labor-associated gene promoters in myometrial cells. by Nawrah Khade et al.

This study has identified some SOX family members with enhanced expression in murine actively contractile myocytes in spontaneous and induced conditions. It could reveal novel roles for SOX4 in expressional regulation of “contractile” genes by luciferase assay. Mmp11 is found up-regulated by SOX4, while Ptgs2 is down-regulated. Though precise mechanisms remain to be elucidated, the study clarify some casts, including SOX4, involve drastic changes in laboring uterine myocytes.

The study obeys all the guidelines

The reviewer generally accepts the data presented and the discussion and conclusion from the data. The reviewer has some concerns below.

Major points

1. Is Fig. 3B the typical blot of SOX4? It does not seem to match with the metric data of C. The quality of this Western analysis is low and better quality blot should be provided.

2. The statistical analysis In Fig.4, there seems to be statistically significant differences. For example, B pG-FosP vs pG-FosP, SOX4, and C pG-Mmp11 vs pG-Mmp11, JUND/FOSL2. Please confirm again these points. The discussion and conclusion (Lines 241-244) may be needed to be modified when there be significant difference.

Minor points are

1. Is ERK2 the adequate marker of constantly expressed proteins by the myocytes? This enzyme can be phosphorylated. And the metric data of ERK blots included only unphosphorylated or the sum of un- and phosphorylated ones? The reviewer requests authors’ notice in the rebuttal or in the text.

2. Indication of significant level of difference In Fig.4B, D is not needed. pG-Gja1P, SOX4 is AB, and pG-Gja1P, JUND/JUND/SOX4 is B. That is OK.

3. Collection of myometrial samples (Line 408-410) Did authors accurately excise out the endometrial and stromal tissues from uterine samples?

4. What is the source of Syrian hamster myometrium (SHM)? There is no description in the text.

5. It should be indicated in the figure legends or somewhere that error bars mean SD.

Reviewer #2: In the current study, the authors investigated the roles of SOX family TFs in myometrial cells, and found that Sox4 is elevated in murine myometrium during term labor and in two mouse models of preterm labor. Using luciferase reporter assay, they also revealed that SOX4 differentially affects labor associated gene promoter activity, in cooperation with AP-1 dimers. This study provides new clues that SOX4 paly a diverse role in regulating gene expression in the laboring myometrium. However, there are several issues to be further addressed.

1. To confirm the effects of SOX4, and more important, to dissect the diverse effects of SOX4 on the different labor associated genes, the mRNA and protein expression of the target genes need to be determined after SOX4 was overexpressed in the cell model.

2. In figure 2A and 2B, the labels in the statistical diagram are not consistent with those in the schematic diagrams on the first lane, which causes confusion.

3. Fig. 3A: why did the author use ERK2 as internal control, which is not a classical housekeeping gene, and may be substantially affected by laboring process?

4. In the previous study, the authors determined the Sox family expression in CD-1 mouse uterus, why didn’t they use the same cell model in the following luciferase reporter experiments, instead of Syrian hamster myometrium (SHM) cell?

6. PLOS authors have the option to publish the peer review history of their article (what does this mean?). If published, this will include your full peer review and any attached files.

Reviewer #1: No

Reviewer #2: **Yes: **Lu Gao

---

## [Author Response · Author response to Decision Letter 0]

30 Nov 2023

Date: Nov 22 2023 11:23AM

To: "Jennifer A. Mitchell" ja.mitchell@utoronto.ca

From: "PLOS ONE" plosone@plos.org

Subject: PLOS ONE Decision: Revision required [PONE-D-23-29498]

PONE-D-23-29498

SOX4 exerts contrasting regulatory effects on labor-associated gene promoters in myometrial cells.

PLOS ONE

Dear Dr. Mitchell,

Thank you for submitting your manuscript to PLOS ONE. After careful consideration, we feel that it has merit but does not fully meet PLOS ONE’s publication criteria as it currently stands. Therefore, we invite you to submit a revised version of the manuscript that addresses the points raised during the review process.

We look forward to receiving your revised manuscript.

Kind regards,

Atsushi Asakura, Ph.D

Academic Editor

PLOS ONE

Journal Requirements:

This work was supported by the Canadian Institutes of Health Research (FRN 173252, held by J.A.M. and O.S.; cihr-irsc.gc.ca). Studentship funding was provided by Ontario Graduate Scholarship (OGS held by N.K.). The funders had no role in study design, data collection and analysis, decision to publish, or preparation of the manuscript.

5. We note that Figures 4A and 4B in your submission contain copyrighted images. All PLOS content is published under the Creative Commons Attribution License (CC BY 4.0), which means that the manuscript, images, and Supporting Information files will be freely available online, and any third party is permitted to access, download, copy, distribute, and use these materials in any way, even commercially, with proper attribution. For more information, see our copyright guidelines: http://journals.plos.org/plosone/s/licenses-and-copyright.

We require you to either present written permission from the copyright holder to publish these figures specifically under the CC BY 4.0 license, or (2) remove the figures from your submission:

a. You may seek permission from the original copyright holder of Figures 4A and 4B to publish the content specifically under the CC BY 4.0 license. 

We have now added the caption at the end of the manuscript (line 596-598).

Reviewers' comments:

Reviewer's Responses to Questions

Comments to the Author

1. Is the manuscript technically sound, and do the data support the conclusions?

Reviewer #1: Partly

Reviewer #2: Partly

2. Has the statistical analysis been performed appropriately and rigorously?

Reviewer #1: I Don't Know

Reviewer #2: Yes

3. Have the authors made all data underlying the findings in their manuscript fully available?

Reviewer #1: Yes

Reviewer #2: Yes

4. Is the manuscript presented in an intelligible fashion and written in standard English?

Reviewer #1: Yes

Reviewer #2: Yes

5. Review Comments to the Author

Reviewer #1: SOX4 exerts contrasting regulatory effects on labor-associated gene promoters in myometrial cells. by Nawrah Khade et al.

This study has identified some SOX family members with enhanced expression in murine actively contractile myocytes in spontaneous and induced conditions. It could reveal novel roles for SOX4 in expressional regulation of “contractile” genes by luciferase assay. Mmp11 is found up-regulated by SOX4, while Ptgs2 is down-regulated. Though precise mechanisms remain to be elucidated, the study clarify some casts, including SOX4, involve drastic changes in laboring uterine myocytes.

The study obeys all the guidelines

The reviewer generally accepts the data presented and the discussion and conclusion from the data. The reviewer has some concerns below.

We would like to thank the reviewer for their assessment of our work.

Major points

1. Is Fig. 3B the typical blot of SOX4? It does not seem to match with the metric data of C. The quality of this Western analysis is low and better quality blot should be provided.

RESPONSE: Figure 3 has now been updated with a higher quality blot of SOX4.

2. The statistical analysis In Fig.4, there seems to be statistically significant differences. For example, B pG-FosP vs pG-FosP, SOX4, and C pG-Mmp11 vs pG-Mmp11, JUND/FOSL2. Please confirm again these points. The discussion and conclusion (Lines 241-244) may be needed to be modified when there be significant difference.

RESPONSE: As seen in the statistical significance below (highlighted), Tukey's multiple comparisons test confirmed that these groups are not statistically significant which is reflected in our figure.



Tukey's multiple comparisons test Mean Diff. 95.00% CI of diff. Below threshold? Summary Adjusted P Value 

pG-FosP vs. pG-FosP, JUND/JUND -0.8934 -2.768 to 0.9811 No ns 0.6129 A-B

pG-FosP vs. pG-FosP, JUND/FOSL2 -9.601 -11.48 to -7.726 Yes **** <0.0001 A-C

pG-FosP vs. pG-FosP, SOX4 -1.470 -3.344 to 0.4047 No ns 0.1620 A-D

pG-FosP vs. pG-FosP, JUND/JUND/SOX4 -5.463 -7.338 to -3.589 Yes **** <0.0001 A-E

pG-FosP vs. pG-FosP, JUND/FOSL2/SOX4 -10.68 -12.55 to -8.806 Yes **** <0.0001 A-F

pG-FosP, JUND/JUND vs. pG-FosP, JUND/FOSL2 -8.707 -10.58 to -6.833 Yes **** <0.0001 B-C

pG-FosP, JUND/JUND vs. pG-FosP, SOX4 -0.5764 -2.451 to 1.298 No ns 0.8979 B-D

pG-FosP, JUND/JUND vs. pG-FosP, JUND/JUND/SOX4 -4.570 -6.444 to -2.695 Yes **** <0.0001 B-E

pG-FosP, JUND/JUND vs. pG-FosP, JUND/FOSL2/SOX4 -9.787 -11.66 to -7.912 Yes **** <0.0001 B-F

pG-FosP, JUND/FOSL2 vs. pG-FosP, SOX4 8.131 6.256 to 10.01 Yes **** <0.0001 C-D

pG-FosP, JUND/FOSL2 vs. pG-FosP, JUND/JUND/SOX4 4.137 2.263 to 6.012 Yes **** <0.0001 C-E

pG-FosP, JUND/FOSL2 vs. pG-FosP, JUND/FOSL2/SOX4 -1.080 -2.954 to 0.7950 No ns 0.4284 C-F

pG-FosP, SOX4 vs. pG-FosP, JUND/JUND/SOX4 -3.993 -5.868 to -2.119 Yes *** 0.0001 D-E

pG-FosP, SOX4 vs. pG-FosP, JUND/FOSL2/SOX4 -9.210 -11.08 to -7.336 Yes **** <0.0001 D-F

pG-FosP, JUND/JUND/SOX4 vs. pG-FosP, JUND/FOSL2/SOX4 -5.217 -7.092 to -3.343 Yes **** <0.0001 E-F

Tukey's multiple comparisons test Mean Diff. 95.00% CI of diff. Below threshold? Summary Adjusted P Value 

pG-Mmp11 vs. pG-Mmp11, JUND/JUND -1.163 -3.777 to 1.451 No ns 0.6740 A-B

pG-Mmp11 vs. pG-Mmp11, JUND/FOSL2 -2.017 -4.632 to 0.5966 No ns 0.1726 A-C

pG-Mmp11 vs. pG-Mmp11, SOX4 -2.289 -4.903 to 0.3254 No ns 0.0997 A-D

pG-Mmp11 vs. pG-Mmp11, JUND/JUND/SOX4 -9.075 -11.69 to -6.461 Yes **** <0.0001 A-E

pG-Mmp11 vs. pG-Mmp11, JUND/FOSL2/SOX4 -9.367 -11.98 to -6.753 Yes **** <0.0001 A-F

pG-Mmp11, JUND/JUND vs. pG-Mmp11, JUND/FOSL2 -0.8545 -3.469 to 1.760 No ns 0.8731 B-C

pG-Mmp11, JUND/JUND vs. pG-Mmp11, SOX4 -1.126 -3.740 to 1.488 No ns 0.7009 B-D

pG-Mmp11, JUND/JUND vs. pG-Mmp11, JUND/JUND/SOX4 -7.912 -10.53 to -5.298 Yes **** <0.0001 B-E

pG-Mmp11, JUND/JUND vs. pG-Mmp11, JUND/FOSL2/SOX4 -8.204 -10.82 to -5.590 Yes **** <0.0001 B-F

pG-Mmp11, JUND/FOSL2 vs. pG-Mmp11, SOX4 -0.2713 -2.885 to 2.343 No ns 0.9991 C-D

pG-Mmp11, JUND/FOSL2 vs. pG-Mmp11, JUND/JUND/SOX4 -7.058 -9.672 to -4.444 Yes **** <0.0001 C-E

pG-Mmp11, JUND/FOSL2 vs. pG-Mmp11, JUND/FOSL2/SOX4 -7.350 -9.964 to -4.735 Yes **** <0.0001 C-F

pG-Mmp11, SOX4 vs. pG-Mmp11, JUND/JUND/SOX4 -6.786 -9.400 to -4.172 Yes **** <0.0001 D-E

pG-Mmp11, SOX4 vs. pG-Mmp11, JUND/FOSL2/SOX4 -7.078 -9.692 to -4.464 Yes **** <0.0001 D-F

pG-Mmp11, JUND/JUND/SOX4 vs. pG-Mmp11, JUND/FOSL2/SOX4 -0.2919 -2.906 to 2.322 No ns 0.9988 E-F

Minor points are

1. Is ERK2 the adequate marker of constantly expressed proteins by the myocytes? This enzyme can be phosphorylated. And the metric data of ERK blots included only unphosphorylated or the sum of un- and phosphorylated ones? The reviewer requests authors’ notice in the rebuttal or in the text.

RESPONSE: Expression levels of ERK1 and ERK2 protein remain unchanged and are therefore not gestationally regulated or affected by the labouring or postpartum processes (PMID: 12372814) and was confirmed by Li and colleagues (PMID: 12388473). The relative protein expression was normalized with ERK2 as it was the most stable protein expressed as previously determined and has been used in multiple recent publications for densitometric analysis using myometrial tissues in the rat (PMID: 17715430, 19602722, 28945005), mouse (PMID: 28945005, 34418351) myometrium throughout gestation and labour, as well as in the human myometrium (PMID: 29042599, 34114342), and primary human myometrial smooth muscle cells (PMID: 35011690). We chose not to use Actin and GAPDH as they are gestationally regulated in the rodent myometrium. For these reasons, we chose to normalize with total ERK2 using the sc-154 antibody as is now reflected in the manuscript.

2. Indication of significant level of difference In Fig.4B, D is not needed. pG-Gja1P, SOX4 is AB, and pG-Gja1P, JUND/JUND/SOX4 is B. That is OK.

RESPONSE: We have now made the appropriate changes to Fig 4.

3. Collection of myometrial samples (Line 408-410) Did authors accurately excise out the endometrial and stromal tissues from uterine samples?

RESPONSE: We previously specified the removal of the decidua (terminology for pregnant endometrium) however we have now added that the uterine stromal tissues were also removed (line 407-409).

The text now reads: “Both uterine horns were bisected longitudinally and dissected away from both pups and placentas and placed in ice-cold PBS. Through mechanical scraping on ice, the decidua, as well as the uterine stromal tissues, were carefully removed from the myometrium tissue.”

4. What is the source of Syrian hamster myometrium (SHM)? There is no description in the text.

RESPONSE: We have added the source of the SHM cells in our text (line 427).

The text now read: “Syrian hamster myometrium (SHM) cells, generously supplied by Professor Oksana Shynlova..”

5. It should be indicated in the figure legends or somewhere that error bars mean SD.

RESPONSE: We have made the appropriate changes in the figure legends (line 188 and 225) to indicate where data displayed use SEM for error bars (animal experiments) or SD (transfection experiments).

Reviewer #2: In the current study, the authors investigated the roles of SOX family TFs in myometrial cells and found that Sox4 is elevated in murine myometrium during term labor and in two mouse models of preterm labor. Using luciferase reporter assay, they also revealed that SOX4 differentially affects labor associated gene promoter activity, in cooperation with AP-1 dimers. This study provides new clues that SOX4 paly a diverse role in regulating gene expression in the laboring myometrium. However, there are several issues to be further addressed.

We would like to thank the reviewer for their assessment of our work.

1. To confirm the effects of SOX4, and more important, to dissect the diverse effects of SOX4 on the different labor associated genes, the mRNA and protein expression of the target genes need to be determined after SOX4 was overexpressed in the cell model.

RESPONSE: We agree that to determine the role of SOX4 in labor-associated gene expression it would be appropriate to overexpress SOX4 in myometrial cells, however, this would best be done in a mouse model to assess the effect on pregnancy. Although the SHM cell model is routinely used for this type of reporter assay, as they are easy to transfect, it would not be the most appropriate situation in which to investigate the role of SOX4 on endogenous gene expression since these are hamster cells and we have been looking at the effect of SOX4 on mouse promoter driven expression. We have been quite careful in the text to ensure that we refer to the effect of SOX4 on promoter driven transcription and not refer to SOX4 as affecting endogenous mRNA or protein levels except in the discussion where we have proposed that this could be occurring.

2. In figure 2A and 2B, the labels in the statistical diagram are not consistent with those in the schematic diagrams on the first lane, which causes confusion.

RESPONSE: We have now made the appropriate modification to Figure 2.

3. Fig. 3A: why did the author use ERK2 as internal control, which is not a classical housekeeping gene, and may be substantially affected by laboring process?

RESPONSE: As stated above in response to reviewer 1:

Expression levels of ERK1 and ERK2 protein remain unchanged and are therefore not gestationally regulated or affected by the labouring or postpartum processes (PMID: 12372814) and was confirmed by Li and colleagues (PMID: 12388473). The relative protein expression was normalized with ERK2 as it was the most stable protein expressed as previously determined and has been used in multiple recent publications for densitometric analysis using myometrial tissues in the rat (PMID: 17715430, 19602722, 28945005), mouse (PMID: 28945005, 34418351) myometrium throughout gestation and labour, as well as in the human myometrium (PMID: 29042599, 34114342), and primary human myometrial smooth muscle cells (PMID: 35011690). We chose not to use Actin and GAPDH as they are gestationally regulated in the rodent myometrium. For these reasons, we chose to normalize with total ERK2 using the sc-154 antibody as is now reflected in the manuscript.

At line 455-456 the text now reads: “Membranes were then stripped and re-probed using an antibody that detects total ERK2 (sc-154) as it has shown to have the most stable protein expression in the myometrium (18,39)"

4. In the previous study, the authors determined the Sox family expression in CD-1 mouse uterus, why didn’t they use the same cell model in the following luciferase reporter experiments, instead of Syrian hamster myometrium (SHM) cell?

RESPONSE: To our knowledge, the only mouse myometrial cell lines are the “SMU1-10” cells (PMID: 9170108) and m-M116 cells (PMID: 8641172) which have not been used in the context of reporter assays. Since none of these cells been transfected in the past, we are not in favour of purchasing this line (cost $6,376.50 USD for the SMU1-10 cells) to determine if they are transfectable. Additionally, our preliminary experiments have shown that transfection efficiencies of primary myometrial cell lines, both human and mouse are very low (~3-5%), which makes the experiment in these lines unfeasible in these cell lines as well. SHM cells have been used extensively as a model system for reporter assays within a myometrial context (PMID: 36595497, PMID: 27220952, PMID: 15618352) and have a high transfection efficiency (see image below of transfection with pmaxGFP). Additionally, to stay consistent with our previously published findings that used SHM cells for MYB and ELF3 in the same context (PMID: 36595497), we chose to use these cells for our assays.

6. PLOS authors have the option to publish the peer review history of their article (what does this mean?). If published, this will include your full peer review and any attached files.

Do you want your identity to be public for this peer review? For information about this choice, including consent withdrawal, please see our Privacy Policy.

Reviewer #1: No

Reviewer #2: Yes: Lu Gao

---

## [Editor Report · Decision Letter 1]

14 Jan 2024

SOX4 exerts contrasting regulatory effects on labor-associated gene promoters in myometrial cells.

PONE-D-23-29498R1

Dear Dr. Mitchell,

We’re pleased to inform you that your manuscript has been judged scientifically suitable for publication and will be formally accepted for publication once it meets all outstanding technical requirements.

Kind regards,

Atsushi Asakura, Ph.D

Academic Editor

PLOS ONE
---

## [Editor Report · Acceptance letter]

1 Apr 2024

PONE-D-23-29498R1 

PLOS ONE

Dear Dr. Mitchell, 

I'm pleased to inform you that your manuscript has been deemed suitable for publication in PLOS ONE. Congratulations! Your manuscript is now being handed over to our production team.

Kind regards, 

on behalf of

Dr. Atsushi Asakura 

Academic Editor

PLOS ONE